# Combined Freeze-Thaw and Chloride Attack Resistance of Concrete Made with Recycled Brick-Concrete Aggregate

**DOI:** 10.3390/ma14237267

**Published:** 2021-11-28

**Authors:** Yao Yu, Jian Wang, Ninghui Wang, Chenjie Wu, Xiaojing Zhang, Dezhi Wang, Zhipeng Ma

**Affiliations:** 1School of Civil & Water Conservancy Engineering, Ningxia University, Yinchuan 750021, China; yuyao2021@nxu.edu.cn (Y.Y.); wangjian2021@nxu.edu.cn (J.W.); wangninghui2021@nxu.edu.cn (N.W.); wuchenjie2021@nxu.edu.cn (C.W.); zhangxiaojing2021@nxu.edu.cn (X.Z.); mazhipeng2021@nxu.edu (Z.M.); 2Ningxia Water-Efficient Irrigation Engineering Research Center, Yinchuan 750021, China; 3Water Resources Engineering Research Center in Modern Agriculture in Arid Regions, Yinchuan 750021, China

**Keywords:** recycled aggregates, fly ash, corrosion, water-soluble chloride content, freeze-thaw

## Abstract

The objective of this study was to investigate the physico-chemical properties of concrete made with recycled brick-concrete aggregate, which was the mixture from waste concrete and waste clay brick in a 7:3 ratio. Specifically, this paper investigated the mechanical properties, freeze-thaw resistance, and distribution of water-soluble chloride ions of concrete containing RBCA and fly ash (FA) against combined freeze-thaw and sodium chloride attack. Concrete containing RBCA replacement of natural coarse aggregate and fly ash replacement of Portland cement was subjected to 45 freeze-thaw cycles containing sodium chloride solution. It was discovered that the mechanical properties and freeze-thaw resistance to sodium chloride attack gradually decreased with increasing RBCA content. At the same time, a replacement level of 15% FA by weight resulted in significant improvements in compressive strength and resistance to combined freeze-thaw and chloride attack. Furthermore, using a replacement of 30% FA by weight markedly improved the resistance to chloride ion penetration of concrete due to the lowest water-soluble chloride content.

## 1. Introduction

As industrial growth and urbanization continue, the cumulative amount of construction solid waste grows rapidly. In the last five years, the utilization rate of construction solid waste in Japan, the Netherlands, Germany, and Denmark has reached 80% or more, whereas, in China, the utilization rate is only around 47.7% [1]. As a result, improving the replacement rate of recycled aggregate and the performance of concrete has become a critical research area to enhance the recycling process of recycled aggregate. At present, the majority of the recycled aggregate utilized domestically in China and internationally is made from waste concrete after crushing, sorting, and screening. Actually, clay brick which is the main material of masonry structures, accounts for a significant portion of building and demolition waste. If broken bricks were widely employed in concrete manufacturing, the shortage of natural sand and stone resources would be alleviated, the recovery efficiency of building solid waste would be increased [2,3].

In general, recycled aggregate degrades the performance of concrete, as demonstrated by a significant number of scholars [4,5,6,7]. The quality of recycled concrete was depended on many factors, including the strength of parent concrete of recycled concrete aggregate (RCA) and the proportion of RCA [8]. Typically, for RCA with higher strength of parent concretes (PC), a lower replacement rate was conducive to the strength of recycled concrete. Kou [9] prepared concrete with RCA derived from multiple PC, including 30, 45, 80, and 100 MPa. The tests indicated that the mechanical properties of the recycled concrete derived from PC of 30 and 45 MPa were fallen 21.1% and 12.6%, while concrete with RAC from another two PC and natural aggregate presented almost the same. Statistical analyses by Ohemeng [10] concluded that recycled aggregate would be unfavorable to the performance of concrete when PC was lower than 50 MPa. As the proportion of RCA increased, the performance of concrete was worse due to amplified porosity [11]. Moghadam [12] conducted mechanical experiments on three concrete mixtures containing 0%, 50%, and 100% by weight of RCA from three types of PC. When the strength of PC was 20, 40, and 80 MPa, the strength ratios of concrete with 50% RCA and 100% RCA were 1.20, 1.11, and 1.08, respectively. The results discovered that the mechanical properties presented a downward trend with increasing RCA while elevating the PC strength of the RCAs can weaken the strength loss rate. In other words, the replacement percentage of recycled aggregate was the main indicator to dominate the mechanical properties of concrete while the strength of PC was constant. Indeed, performance degradation of recycled concrete was due to the presence of two additional interfacial transition zones (ITZ) with low particle density. One of them existed between natural aggregate and old mortar adhered to the recycled aggregate, and the other existed between natural mortar and old mortar [13,14]. The microstructure of ITZ was observed by scanning electron microscope (SEM), the concentration of calcium hydroxide crystal was higher, and the porosity was larger [15]. The porosity of the mortar matrix is about 1/3 of the maximum porosity of ITZ [16]. As the residual old mortar was difficult to be removed completely, the higher volume fraction of aggregate was, the larger the weak area would be [17]; therefore, the replacement rate affected the macro performance of concrete.

In real environments, structural failure is not only caused by inadequate strength but also caused by a combination between mechanical and durability of deleterious issues. Concrete durability was adversely influenced by many physical effects, including wear of frictional surface, cracking on account of pore solution crystallization, faced with the atrocious circumstance in temperature (frost or heat action, for instance), and detrimental chemical effects, including lixiviating of the cement paste because of acidic solutions [18] and expansive substances by sulfate attack [19,20]. In comparison, corrosion of reinforcement by the chlorine ion in concrete structures [21] was one of the biggest and the most serious factors in saline-alkali soil, areas covered with deicing salts, and coastal environments. Under the combination of chlorine corrosion and the freeze-thaw cycle, a more complex physical and chemical process occurs where high free chloride ions would partially destroy the passive film of reinforcement, and then the iron matrix would be exposed. The corrosive battery was formed due to the potential difference between iron and passive film. More severely, the temperature fluctuation further speeds up chloride diffusion. The deterioration rate of concrete is controlled by the migration of water in concrete. On this account, the parameter that the influence of the permeability of concrete on durability should be evaluated. Fly ash, as an industrial by-product, was widely used to improve concrete performance by virtue of its value attributes such as micro-aggregate filling and pozzolanic effect [22,23]. The micro-aggregate filling effect reduced porosity and the pozzolanic effect consumed Ca(OH)_2_, and then ITZ is enhanced. Finally, concrete performance was enhanced wholly because the permeability of the constituent materials determined the permeability of concrete. Kou et al. [24] stated that fly ash in proper proportion could improve the frost resistance of recycled concrete and enhance its chloride penetration resistance to some extent. They tested recycled concrete with 0, 25%, and 35% Class F fly ash and pointed out that incorporating 25–35% fly ash was a practical way to improve the chloride permeability resistance at the water-binder ratio of 0.45 and 0.55, which minimized the total charge passed of recycled concrete at 28 and 90 days. Moreover, Kurda [25] extracted the results of 655 concrete incorporating coarse RCA and FA, at least 28 days, from many papers. It was found that the concrete with 20% fly ash exhibited a better performance compared to the benchmark concrete. Other researchers [26,27] also confirmed through tests such as the RCM and electric quantity methods that fly ash can improve concrete impermeability, but higher levels of fly ash incorporation can be detrimental to chloride penetration resistance of recycled concrete as a result of leading to higher porosity.

A good deal of research on the mechanical properties and durability of concrete made with waste concrete aggregates may exist. However, research regarding the deterioration of concrete composed of recycled brick-concrete aggregate (RBCA) is limited in the literature and mainly focuses on the mechanical properties [28,29,30,31]. Generally, the crushed clay brick aggregate absorbs more water than nature aggregate or recycled concrete aggregate, and its porosity and crushing index are significantly greater. Poon [14] mixed recycled brick aggregate and recycled concrete aggregate together, and the test results indicated that the maximum dry density was lesser and the optimum moisture content larger than RCA solely. Similarly, the mechanical properties of recycled concretes containing brick were also affected by the replacement rate. Boukour manufactured mortars with 2.5%, 5%, 7.5%, and 10% brick waste of natural coarse aggregate, and there was a gradual downward trend in the 28 days compressive and flexural strengths [32]. However, in terms of durability, there were two opposite effects of aggregate size and volume fraction on the migration coefficient of chloride ions. The positive effects included dilution and deformation effect, which reduced the migration coefficient, and the negative effects included transition zone and seepage, which increased the value [16]. The aggregate size and porosity are easy to measure. However, when the shape of concrete aggregate inclusion is not standard, the information of chloride ion migration coefficient in the interface transition zone is very little, and it is difficult to test the interface transition zone alone. Lutz [33] estimated the effective macroscopic conductivity of the medium by the Maxwell homogenization method, which calculated the effective conductivity from a single-inclusion problem, and verified that the conductivity decreased due to non-conductive inclusions while increased due to ITZ. Until now, it has been difficult to unite the physical markers of recycled brick, and dispersion in the findings persists. While there have been some advancements in the mechanical properties of recycled brick-concrete aggregate (RBCA) concrete, there is a gap in the knowledge needed to evaluate the durability of RBCA concrete. Given the vast number of discarded bricks in construction solid waste, there is significant potential for study into the performance of recycled concrete using RBCA.

As a result, in this study, the recycled brick–concrete aggregate (RBCA) was made from waste concrete and waste clay bricks in a ratio of 7:3. Mechanical properties and durability (frost resistance and permeability) of concrete with RBCA were investigated, and the influence of RBCA and FA under the combined action of cyclic freeze-thaw and chloride attack. Furthermore, the phase compositions of recycled concrete were determined using microscopy and X-ray diffraction.

## 2. Experimental Program

### 2.1. Experimental Materials

#### 2.1.1. Cement

An amount of 42.5R Ordinary Portland cement (OPC), in conformity with Chinese standard GB175-2007 [34], which resemblance to EN197-1:2009, was used to make concrete specimens. The composition and physical properties of OPC are illustrated in Table 1.

#### 2.1.2. Natural Aggregates

Classification of particle size was based on standard ASTM C125. Washed river sand in the range of 4.75 mm (No. 4) and 0.15 mm (No. 100) was used as the fine aggregate. Crushed limestone in the range of 31.5 mm ($1\frac{1}{4}$in.) and 4.75 mm (No. 4) was used as the coarse aggregate.

#### 2.1.3. Recycled Aggregates

After removing steel reinforcement bars from waste concrete floor slabs, they were crushed in a jaw crusher, screened, and washed to obtain a continuous grain size range of 4.75–31.5 mm, and finally mixed with recycled brick aggregate at a 7:3 ratio. The compressive strength of parent concretes was 40.7 MPa. In this study, the compound aggregate was named RBCA, and recycled concrete aggregate and recycled brick aggregate solely was named RCA and RBA; RBAC, RAC, and RBA had the same particle gradation. Physical characteristics test methods in accordance with Chinese standard JGJ 52-2006 [35]. Table 2 and Figure 1 present the physical properties and the grading chart of recycled aggregate, respectively. Additionally, the crushing value index of RBCA was 20.3%, and the results were classified as type III according to Chinese standard GB/T 25177-2010 [36].

#### 2.1.4. Other Materials

The polycarboxylate superplasticizer was used in the concrete mixtures to maintain the concrete slump of 90–100 mm, and the slump loss in one hour was 40–50 mm. Anhydrous sodium chloride with 99.5% purity provided the raw material for chloride solutions. The Class F fly ash (FA) was used as supplementary cementitious material.

### 2.2. Mix Proportion

Concrete was manufactured with a water-binder ratio of 0.4, the binder material of 450 kg/m^3^, and a sand ratio of 0.34. There were two different percentages of RBCA (i.e., 80 and 100 wt.% of natural coarse aggregates). The RBCA was made of waste concrete and waste clay bricks in a ratio of 7:3. There were three different percentages of FA (i.e., 0, 15, and 30 wt.% of OPC). The actual concrete mix proportions are exhibited in Table 3.

### 2.3. Experimental Methods

The preparation process of specimens complied with the standard ASMT C192. Add coarse aggregate, partial water, and the solution of the polycarboxylate superplasticizer into the mixer. Then, start the mixer and add cementitious material, fine aggregate, and remaining water. Mix in the mixer for 3 min, rest for 3 min, and mix for another 2 min to mix them evenly. Prismatic specimens 100 mm × 100 mm × 400 mm and cube specimens 100 mm in size were cast in plastic molds, removed after about 24 h, and cured for 28 days at 20 ± 2 °C with a relative humidity of at least 95%. Prismatic specimens were used to assess frost resistance, while cube specimens tested compressive strength. Axial compressive strength tests were performed in conformity to ASMT C39, and test results were determined to be the mean values of every 3 specimens.

#### 2.3.1. Freeze-Thaw Cycles in NaCl Solution

The rapid freeze-thaw process was in accordance with ASTM C666. Prismatic specimens of 100 mm × 100 mm × 400 mm were cured for 28 days before being immersed at 1 mm depth in water for 7 days. Five surfaces were sealed with epoxy resin glue (except for one side of 100 × 400 mm^2^) to ensure chlorine penetrates from the reserved erosion surface when immersed in saline solution (Figure 2a). A total of 45 freeze-thaw cycles were performed, with each cycle lasting 6 h (3 h frozen at −18 ± 2 °C, and 3 h thawing at 5 ± 2 °C). Mass and transverse frequency were determined for specimens after 15 and 45 freeze-thaw cycles using an electronic scale and a dynamic elastic modulus measurement instrument (Figure 3), respectively. The mass-loss rate (MLR) and the relative dynamic elastic modulus (RDEM) were calculated via Equations (1) and (2).
(1)$$MLR=\frac{m_{0}-m_{n}}{m_{0}}\times 100$$
(2)$$RDEM=\frac{f_{n}^{2}}{f_{0}^{2}}\times 100$$
where *MLR* is the mass loss rate (%); *m_0_* is the initial mass (g); *m_n_* is the mass of a concrete prismatic specimen that underwent *n* freeze-thaw cycles (g). *RDEM* in % is the relative dynamic elastic modulus of concrete; *f*_0_ is the initial transverse frequency (Hz); *f_n_* is the transverse frequency of a prismatic concrete specimen that underwent *n* freeze-thaw cycles (Hz).

#### 2.3.2. Water-Soluble Chloride Test

Drilled concrete vertically inward along the erosion surface by layer as described in Test method ASTM C42 and collected concrete powder samples. Referring to ACI 318, the concrete protective layer thickness is concentrated in 10–50 mm; it is 75 mm only in rare cases. Therefore, the chloride ion concentration was focused on in the range of 0–50 mm from the erosion surface.

The sampling diagram is depicted in Figure 2b, and the water-soluble chloride test procedure was in accordance with ASTM C1218 (as shown in Figure 4). Samples were soaked in distilled water and stood for 24 h after vibrating for 5 min to extract chloride ions. Then 2 drops of phenolphthalein were used as a color indicator, and it turned red because the aqueous solution was alkaline. Subsequently, dilute sulfuric acid was added until the solution was colorless. In the end, 10 drops of K_2_CrO_4_ solution were an indicator added to record the consumption of AgNO_3_ when peach precipitation formed. The chemical reactions are shown in Equations (3) and (4), respectively. The silver ion of AgNO_3_ solution reacted with the chloride ion to form silver chloride, and then excess silver ion interacted with chromate ion of K_2_CrO_4_ solution to form peach precipitation. Chloride ion content was measured by recording the consumption of the AgNO_3_ solution. The water-soluble chloride content (WSCC) was calculated via Equation (5).
(3)$$Ag^{+}+Cl^{-}\to AgCl↓$$
(4)$$2Ag^{+}+CrO_{4}^{2}{}^{-}\to Ag_{2}CrO_{4}↓$$
(5)$$WSCC=\frac{\left(C_{{\mathrm{AgNO}}_{3}}V\times 0.03545\right)}{G\times \frac{V\prime }{V}}\times 100$$
where *WSCC* is the water-soluble chloride content of concrete (%); C_AgNO3_ in 0.02 mol/L is the concentration of AgNO_3_ prepared; V is the AgNO_3_’s consumption of titrations (mL); V′ is the amount of distilled water was used to dissolve the powdery sample.

## 3. Results and Discussion

### 3.1. Compressive Strength of Recycled Concrete

#### 3.1.1. Compressive Strength and RBCA Content

The compressive strength of concrete with different recycled aggregate content at 28 days depicted in Figure 5 shows that the R100 series concrete was weaker in compression than the R80 series concrete across all fly ash contents. When the content of RBCA was increased from 80% to 100%, the compressive strength of concrete with 0%, 15%, and 30% of FA dropped by 2.60%, 7.28%, and 2.18%, respectively. The results indicated a negative effect between the content of RBCA and mechanical properties that the 28-day compressive strength of concrete reduced with an increase in RBCA replacement.

Some differences emerged, by comparison, the values of the R80 and R100 series in Figure 5. That R80 series concrete had a margin of increase of 2.2–7.9% than R100 series concrete in the case of the same mix proportion. Distinctly, this gap could attribute to the volume fraction of recycled aggregate. The extra added brick aggregate offered a mixed effect of positive and negative superimposition. On the one hand, recycled brick aggregate had a lower compressive strength and a higher crushing index than recycled concrete aggregate, so the compressive strength of the R100 series was reduced to some extent [30,31,37]. On the other hand, the brick aggregate had a high porosity and water absorption capacity (18%), which was three times than RCA, allowing it to absorb an amount of mixing water from the mortar mixture early on and slowly release it later. Due to the absorption and desorption properties of the brick aggregate, it can act as a source of curing water for the concrete later. Moisture aided in the further hydration of the cementitious materials, ensuring homogeneous curing within the concrete mass and thus increasing the compressive strength of the mortar [38]. In addition, the actual water-binder ratio of concrete with RBCA below the designed value results in strength was relatively high.

#### 3.1.2. Compressive Strength and FA Content

Trends on the strength curve of concrete with different mix ratios exhibited the same (Figure 6). When the content of FA increased from 0%, 15% to 30%, these curves went up and then down. Specifically, 15% FA generated strength growth by around 7% and 2% under R80 and R100 series, respectively. However, the strength for the concrete produced using 30% FA compared to that of corresponding concrete no fly ash had an average shrinkage of 2.4%. This phenomenon was attributed to micro-aggregate filling and the pozzolanic effects of fly ash [20,21]. Nevertheless, a high volume of fly ash incorporation magnified total porosity result in density reduction.

### 3.2. The Mass Loss Rate (MLR)

As shown in Figure 7, MLR of concrete specimens containing RBCA was performed using 45 freeze-thaw cycles in a 3.0% by weight NaCl solution (3%NS-45FT).

The MLR increased as the number of freeze-thaw cycles and the amount of RBCA in the concrete grew, and its value directly represented the extent of concrete spalling. In the case of R100, the MLR of concrete containing 0, 15%, and 30% FA had increased to 6.4%, 5.6%, and 6.2% of their initial value in the sequence, respectively, whereas in the case of R80, the counterpart values were 6%, 5.1%, and 5.7%, respectively. According to the results, 15% FA reduced MLR to a minimum of 5.1% after 45 freeze-thaw cycles in a 3.0% NaCl solution. Additionally, the MLR of concrete without FA was somewhat greater than concrete containing 30% FA.

### 3.3. The Relative Dynamic Elastic Modulus (RDEM)

Figure 8 illustrates the decline in RDEM of concrete specimens exposed to 3% NS-45FT. All RDEMs exhibited a declining trend, and their amplitudes ranged from 9.58% to 46.92% of their starting value. Furthermore, R100-FA0 had the greatest reduction in RDEM, with a noticeably steep curve and a total RDEM decrement of 53.08%, followed by R100-FA15 and R80-FA15.

In the instance of the R100 series, RDEM grew as fly ash content increased, while the R80 series curves pattern remained consistent. The vertical distance between two curves in concrete without FA (Figure 8b) shows that the RDEM of the concrete was the most responsive to RBCA concentration. In the case of concrete containing 30% FA (Figure 8d), the R100 series had a higher RDEM value than the R80 series, which was in contrast to the behavior of concrete without FA or with 15% FA. As a result, a replacement level of 15–30% FA was favorable to the resilience of concrete subjected to the combined freeze-thaw and salt chloride solution attack. Furthermore, the RDEM value of concrete made with varying RBCA content performs differently depending on the amount of FA.

### 3.4. Water-Soluble Chloride Content (WSCC)

#### 3.4.1. WSCC and RBCA Content

Figure 9 and Figure 10 show the WSCC of concrete samples subjected to 3% NS-45FT. The WSCC of specimens showed a downward trend with increasing distance from the erosion surface, stabilizing at a depth of 30–40 mm, as shown in Figure 9. Even with the variable FA content, the WSCC of the R100 series was typically greater than the R80 series for the same depth. The WSCC curve for concrete with a varied amount of RBCA (i.e., 80 and 100 wt.%) displayed a more significant divergence when concrete contained 15% FA.

The surface WSCC value was larger than the interior value, which stabilized at a depth of 30–40 mm. That was due to a superficial sample in contact with the solution directly. In the early stage, chloride ion penetration in the concrete specimens was dominated by capillary absorption, and the penetration rate was fast [39]. The difference in chloride ion concentration between the internal specimen solution and the immersion solution narrowed as the specimen became saturated, and chloride ion penetration became dominated by diffusion, with a relatively slow penetration rate. However, because areas near the erosion surface were always saturated first, the WSCC in the shallow layer of the specimen was high, while the concentration in the deeper layer was low, and the difference was not significant.

#### 3.4.2. WSCC and FA Content

Figure 11 shows the effect of fly ash content on WSCC after 3%NS-45FT. It demonstrates that WSCC with RBCA dropped dramatically in response to increasing the FA dosage and that 30% FA greatly enhanced resistance to chloride-ion penetration compared to 15% FA.

### 3.5. SEM and XRD

SEM microscopy was used to examine the microstructure of concrete containing RBCA. After undergoing a 3%NS-45FT treatment (Figure 12), the concrete specimens became rather loose, with large holes and noticeable fissures visible.

Differences in microstructure were also detected (see Figure 12a–d), with the concrete specimen containing 100% RBCA having larger pores and more cracks. Through repeated freeze-thaw cycles, those cracks expanded and interacted with the pores and interfacial transition zones, providing pathways for ions to permeate the internal structure from the external environment.

After 45 freeze-thaw cycles, concrete specimens containing 80% RBCA had a denser matrix than those containing 100%, but specimens containing 15% FA were inferior to those containing 30% FA. This suggested that concrete with 15% FA was more susceptible to chloride attack, whereas concrete with 30% FA exhibited stronger chloride ion penetration resistance. Generally, microscopic characteristics matched macroscopic examination results (WSCC test).

X-ray Diffraction (XRD) patterns were used to determine the phase composition of concrete containing RBCA that was subjected to sodium chloride solutions during freezing-thawing cycles. As presented in Figure 13, the samples all had similar diffraction peaks in terms of type and position, indicating no new phases formed in any of the specimens. The distinctive peak of SiO_2_, originating from sand, was the most prominent, followed by Ca(OH)_2_, AFt (3CaO·Al_2_O_3_ CaSO_4_·32H_2_O), and Friedel’s salt (3CaO·Al_2_O_3_·CaCl_2_·10H_2_O, abbreviated as Fs). The presence of Friedel’s salt in the XRD graph suggested that the chloride salt had penetrated the matrix and reacted with the AFm phase [40,41]. Moreover, FA reacted with the Ca(OH)_2_ phase to form C-S-H, a reaction that appeared to be related to the chloride ion’s physical adsorption capacity [42,43,44].

## 4. Conclusions

Concrete containing RBCA and FA were evaluated comprehensively after being subjected to a combined 3% sodium chloride solution and 45 freeze-thaw cycle regime (3%NS-45FT). Based on the compressive strength, mass loss rate (MLR), relative dynamic elastic modulus (RDEM), water-soluble chloride content (WSCC), SEM, and XRD, the following conclusions can be made:The replacement of virgin aggregates with RBCA composed of 30% recycled brick aggregate and 70% recycled concrete aggregate led to a decrease in 28 days compressive strength. The R80 series concrete had a margin of increase of 2.2–7.9% than that of R100 series concrete. The best dosage of FA in combination with recycled aggregate was discovered to be 15%;When the RBCA content increased from 80% to 100%, concrete specimens were susceptible to more damage based on the increased MLR and sharper reduction in RDEM. The concrete MLR was lowest with 15% FA, while RDEM loss was lowest with 30% FA. The addition of 15% FA improved the resistance of concrete following 3% NS-45FT, while the RDMA of various RBCA concrete varied according to the amount of FA;All WSCC followed the same trend, decreasing within 30 mm of the sodium chloride-eroded surface, recovering slightly, and reaching a constant value in the area beyond 30 mm. The chloride penetration resistance of concrete containing 30% FA was the best, owing to the concrete’s low WSCC value.

## Figures and Tables

**Figure 1 materials-14-07267-f001:**
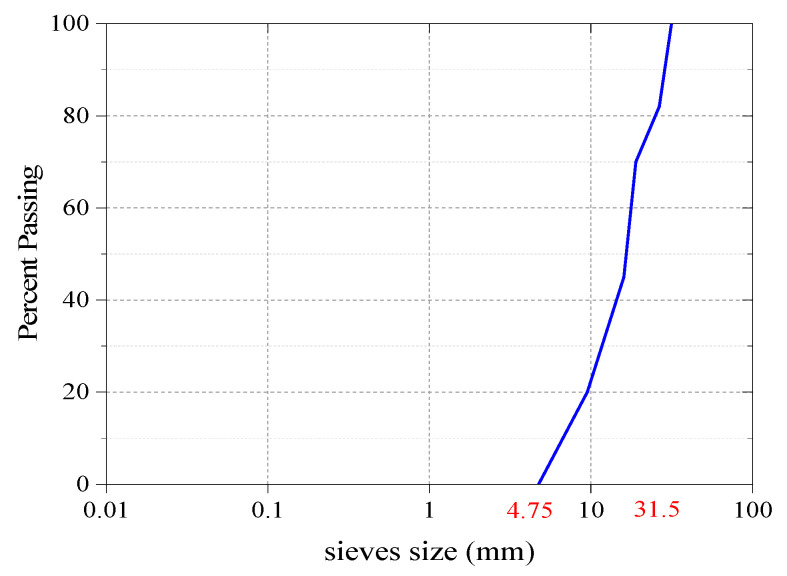
Grading chart of recycled aggregate.

**Figure 2 materials-14-07267-f002:**
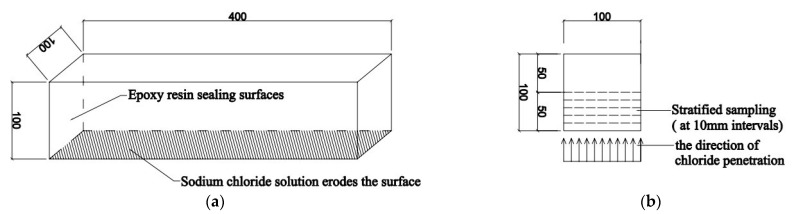
Schematic diagram of sampling for water-soluble chloride content test. (**a**) Schematic of test block for salt-freeze test. (**b**) Cross-section view of stratified sampling.

**Figure 3 materials-14-07267-f003:**
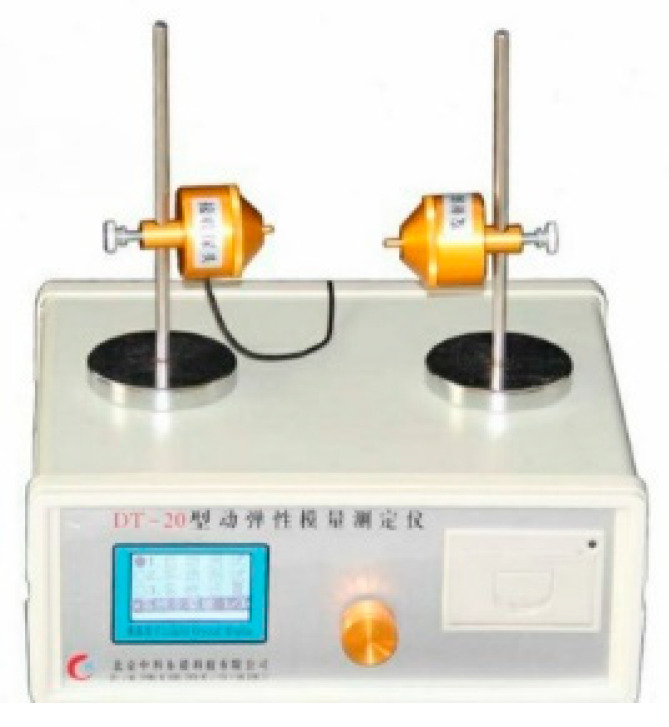
DT-20 Dynamic Elastic Modulus Measurement instrument.

**Figure 4 materials-14-07267-f004:**
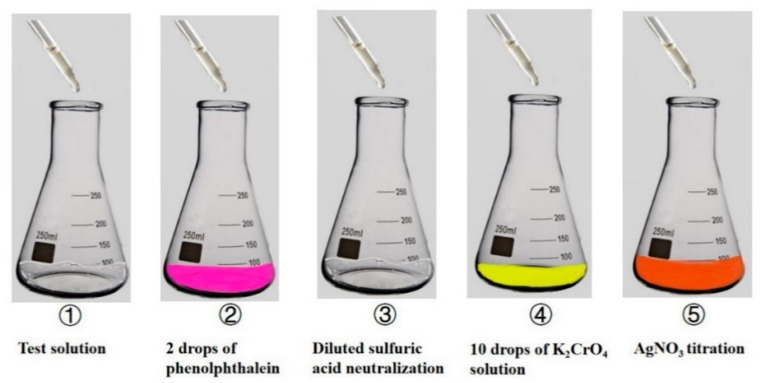
Schematic diagram of the procedure for water-soluble chloride content test.

**Figure 5 materials-14-07267-f005:**
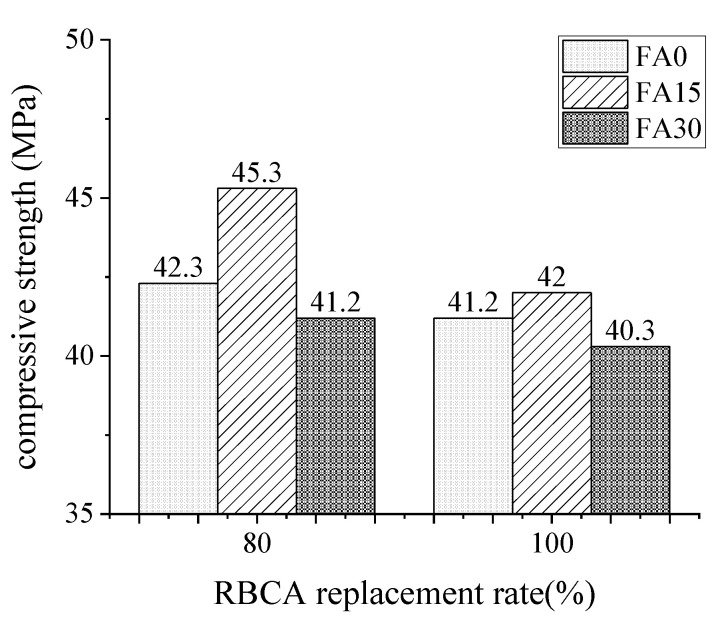
Twenty-eight-day compressive strength of concrete with different recycled aggregate content.

**Figure 6 materials-14-07267-f006:**
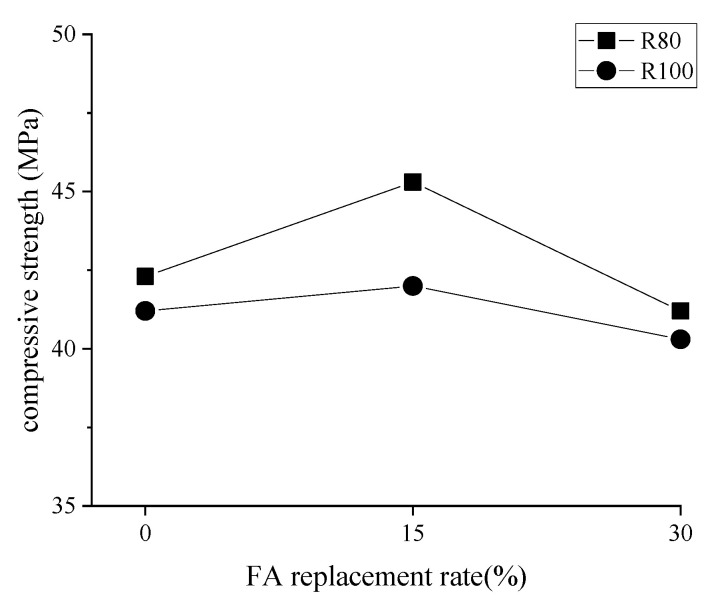
Twenty-eight-day compressive strength in different FA content.

**Figure 7 materials-14-07267-f007:**
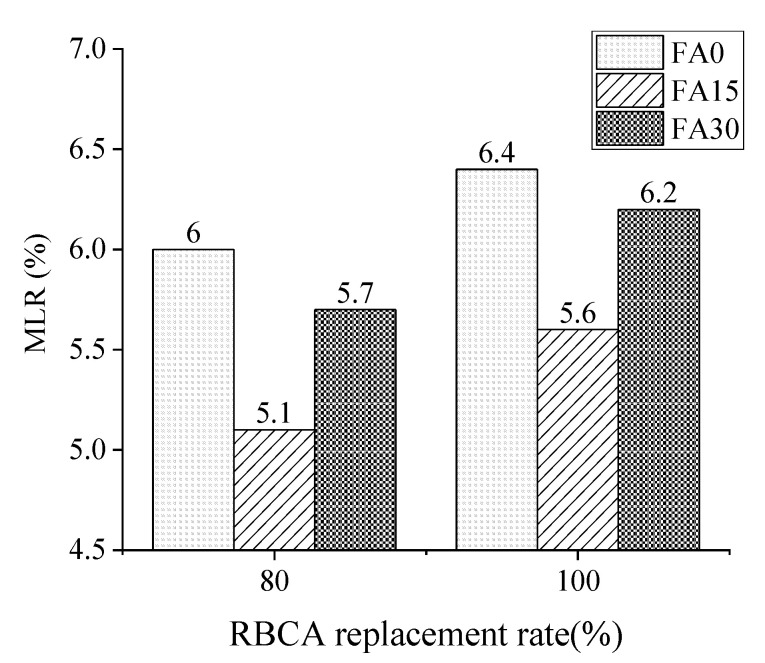
Multifactor influence on the mass loss rate of 3%NS-45FT.

**Figure 8 materials-14-07267-f008:**
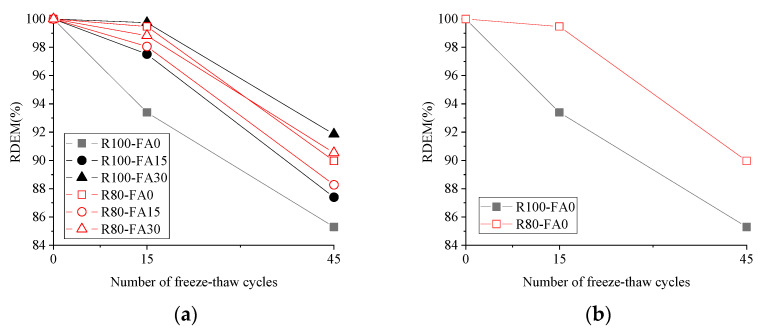
The RDEM of concrete exposed to 3%NS-45FT. (**a**) All, (**b**) FA0, (**c**) FA15, (**d**) FA30, (**e**) R100, (**f**) R80. Note: *R* is for recycled brick-concrete aggregate; *FA* is for fly ash; numbers are the value of replacement rate (%).

**Figure 9 materials-14-07267-f009:**
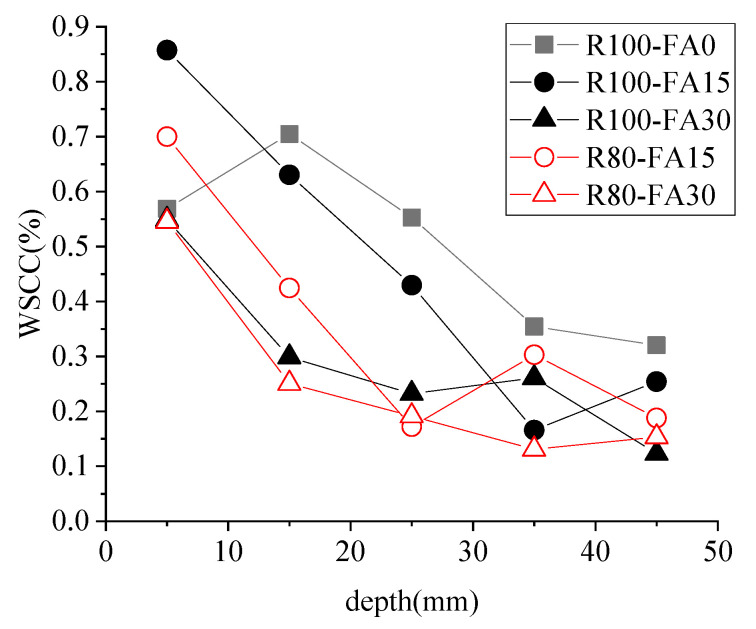
WSCC of concrete samples after 3%NS-45FT.

**Figure 10 materials-14-07267-f010:**
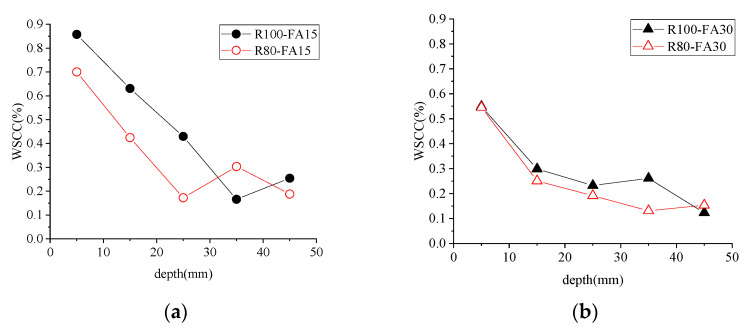
Influence of aggregate replacement rate on WSCC. (**a**) FA15, (**b**) FA30. Note: *FA* is for fly ash; numbers are the value of replacement rate (%).

**Figure 11 materials-14-07267-f011:**
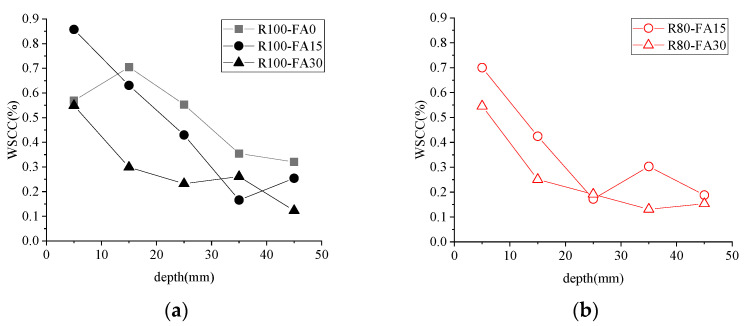
Influence of fly ash content on WSCC. (**a**) R100, (**b**) R80. Note: R is for recycled brick-concrete aggregate; numbers are the value of replacement rate (%).

**Figure 12 materials-14-07267-f012:**
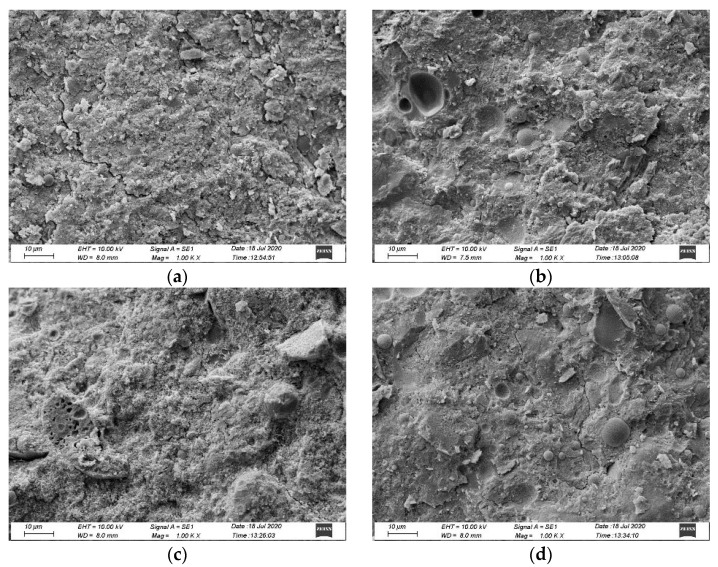
SEM image of specimens with RBCA after 3%NS-45FT (1000×). (**a**) R100-FA15, (**b**) R100-FA30, (**c**) R80-FA15, (**d**) R80-FA30. Note: *R* is for recycled brick-concrete aggregate; *FA* is for fly ash; numbers are the value of replacement rate (%).

**Figure 13 materials-14-07267-f013:**
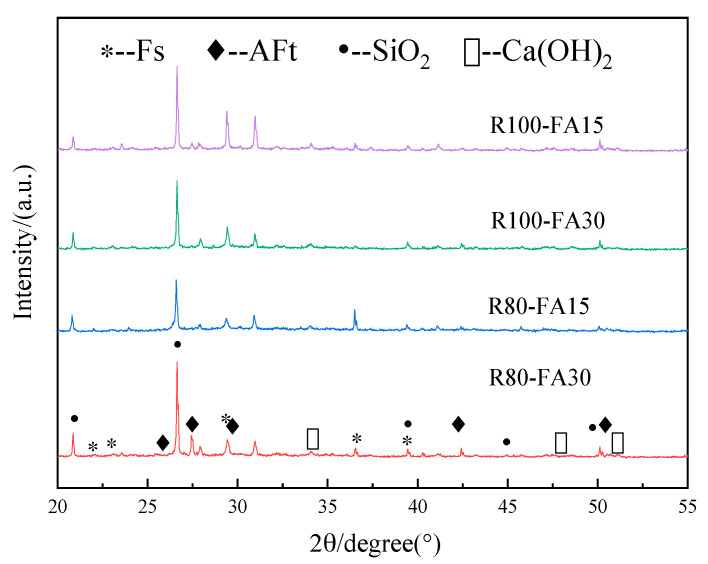
XRD patterns of specimens exposed to 3%NS-45FT.

**Table 1 materials-14-07267-t001:** Composition and physical properties of OPC.

SiO_2_ (%)	Al_2_O_3_ (%)	Fe_2_O_3_ (%)	CaO (%)	MgO (%)	SO_3_ (%)	Specific Surface Area (m^2^/kg)	Initial Setting Time (min)	Final Setting Time (min)	Soundness
14.1	5.19	2.17	43.5	5.00	2.93	348	195	255	qualified

**Table 2 materials-14-07267-t002:** Physical characteristics of recycled aggregates.

No.	Aggregate Sources	Proportion (%)	Water Absorption (%)	Apparent Density (kg/m^3^)	Bulk Density (kg/m^3^)	Crushing Value Index (%)
RBCA	Waste concrete	70	12.8	2302	1020	20.3
	Waste brick	30				
RCA	Waste concrete	100	6.1	2409	1120	15.9
RBA	Waste brick	100	18.0	2051	858	25.5

**Table 3 materials-14-07267-t003:** Mix proportions of recycled concretes (kg/m^3^).

No.	Coarse Aggregate		Natural Sand	Binding Material		Water
	RBCA	Natural Gravel		OPC	FA	
R100-FA0	1201	0	619	450.0	0	180
R100-FA15	1201	0	619	382.5	67.5	180
R100-FA30	1201	0	619	315.0	135	180
R80-FA0	960.8	240.2	619	450.0	0	180
R80-FA15	960.8	240.2	619	382.5	67.5	180
R80-FA30	960.8	240.2	619	315.0	135	180

## Data Availability

The data presented in this study are available on request from the corresponding author.

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
