# Peer review of "Combined Freeze-Thaw and Chloride Attack Resistance of Concrete Made with Recycled Brick-Concrete Aggregate"

_materials, 2021, doi:10.3390/ma14237267_

Round 1
Reviewer 1 Report
This work deals with the study of the physic-mechanical properties of concrete containing different contents of recycled brick-concrete aggregate (RBCA) and (Fly Ash) FA after being subjected to a combined sodium chloride solution and freeze-thaw cycle regimes.
The manuscript id well organized and well written. The objective and the relevance of the contribution to scientific knowledge are clear. The bibliography is updated, since more than 70% of the cited publications are from the last 5 years.
The information on the experimental campaign is complete and the obtained results are comprehensively discussed.
Some errors in references to tables and figures have been found on all pages.
The reviewer thinks this manuscript must be accepted for publications in the Journal Materials.
Author Response
The authors greatly appreciate the reviewer’s support in describing this study as “the paper is well organized and well written…” and praising that “the objective and the relevance of the contribution to scientific knowledge are clear.”.
Errors in references to tables and figures have been corrected on all pages. Here, we attach the revised manuscript, in the formats of both PDF and MS Word, for your approval. A revised manuscript with the correction sections blue marked is attached as the supplemental material and for easy check/editing purpose. We try our best to improve the manuscript and made some changes in the manuscript. We appreciate the Reviewers’ warm work earnestly and hope that the correction will meet with approval.

Reviewer 2 Report
I refer to the uploaded document for my review comments

Author Response
Reviewer #2
Thank you very much for your letter and the referees’ reports. Based on your comment and request, we have made an extensive modification to the original manuscript. Here, we attach the revised manuscript, in the formats of both PDF and MS Word, for your approval. A revised manuscript with the correction sections blue marked is attached as the supplemental material and for easy check/editing purpose. We try our best to improve the manuscript and made some changes in the manuscript. We appreciate the Reviewers’ warm work earnestly and hope that the correction will meet with approval.
Comment 1: the lack of a critical review.
Response 1: The authors have updated the list of references cited. We have rewritten Section 1: Introduction according to the reviewer’s suggestion. The newly cited literatures are mainly from Cement and Concrete Research, Construction and Building Materials, Materials and Structures. We reorganized and summarized these cited references and supplement the corresponding experimental background. In the introduction section, recycled aggregate referring to the research significance, the degradation of mechanical properties, and the durability of concrete were discussed successively. Additionally, we added the influence mechanism of the interface transition zone and the effect of fly ash (Page 2 to 3).
Comment 2: a lack of a mix composition of the different mixes.
Response 2: A total of 6 groups of brick aggregate recycled concrete were studied in this study. Concrete mix proportions were two different percentages of RBCA (i.e. of 80 and 100 wt.% of natural coarse aggregates) and three different percentages of FA (i.e. of 0, 15, and 30 wt.% of OPC). We tested the compressive strength, mass loss rate, relative dynamic elastic modulus. In addition, the water-soluble chloride content of concrete at different depths was tested. We summarized and analyzed the test data again. It has been included in Table 3: Mix proportions of recycled concretes and in Section: 3. Results and discussion (Refer Page 6 and Page 8 of the revised manuscript, respectively).
Comment 3: not innovate and sufficient research.
Response 3: We would like to take the opportunity to highlight the comprehensiveness and originality of this study. First of all, a large number of reinforced concrete structures are severely deteriorated, mainly because of the chloride attack and freeze-thaw damage, and research on the deterioration is a matter of great urgency. Secondly, few studies on the property of recycled concrete with fly ash and recycled aggregate containing clay brick under chloride attack exposed to freeze-thaw cycles have been conducted. Thirdly, we studied the relationship between recycled brick-concrete aggregate or fly ash and the resistance of concrete to freeze-thaw, which was different from the previous studies focusing on recycled aggregate only derived from waste concrete aggregates. Finally, chloride transport properties of concrete with RBCA were studied. All water-soluble chloride content (WSCC) followed the same trend, decreasing within 30 mm of the sodium chloride-eroded surface, recovering slightly, and reaching a constant value in the area beyond 30 mm. As for concretes with 30% FA had a positive effect owing to the concrete’s low WSCC value.
Comment 4: no significant improvement in the index.
Response 4: This paper assessed the influence of recycled aggregates and fly ash under the combined action of cyclic freeze-thaw and chloride attack. Additionally, chloride transport properties of concrete with recycled aggregate were studied. There was a total of 6 concrete mixtures in the revised draft. We found similarities and differences in performance indexes such as compressive strength, mass loss rate, relative dynamic modulus of elasticity, and content of water-soluble chloride ions at different depths. It was discovered that the mechanical properties and freeze-thaw resistance to sodium chloride attack gradually decreased with increasing recycled aggregate content. With the increase of fly ash content, concrete properties were first enhanced and then inhibited. Furthermore, water-soluble chloride content reached a constant value in the area beyond 30 mm. We are trying to explore and hope to achieve good indicators to reflect the improvement effect.
Comment 5: physicochemical.
Response 5: Physico-chemical. It has been revised in Abstract and indicated in blue color (Refer Page 1 of the revised manuscript).
Comment 6: As industrial growth and urbanization continue, the cumulative amount of construction solid waste grows exponentially, yet is largely overlooked.
Response 6: We change it as "As industrial growth and urbanization continue, the cumulative amount of construction solid waste grows rapidly". It has been revised in Paragraph 1of Section 1: Introduction (Page 1).
Comment 7: What is actually meaning by brick blocks? Do you mean masonry?
Response 7: We are very sorry for our ambiguous expression. We have added explanations that “Actually, clay brick which is the main material of masonry structures accounts for a significant portion of building and demolition waste.”. It has been revised in Section 1: Introduction (Page 1).
Comment 8: References [4-12] on the source of citations.
Response 8: We have rewritten this part according to the reviewer’s suggestion. Newly cited literature was from Cement and Concrete Research, Construction and Building Materials, Materials and Structures. It has been revised in Section 1: Introduction (Refer Page 1 to 2).
Comment 9: Damage caused purely by inadequate strength is uncommon in real environments because structural failure depends on a combination of deleterious issue.
Response 9: We changed a more precise way of saying “In real environments, structural failure is not only caused by inadequate strength but also caused by a combination between mechanical and durability of deleterious issues.” It has been revised at the appropriate place in Section 1: Introduction (Page 2).
Comment 10: Which damage mechanisms occur the most?
Response 10: We have summarized the items causing durability damage, but we have not identified the most serious ones, according to the Reviewer’s suggestion, this part was changed to…” In comparison, corrosion of reinforcement by the chlorine ion in concrete structures [21] was one of the biggest and the most serious factors in saline-alkali soil, areas covered with deicing salts, and coastal environments.” Furthermore, the mechanism of steel corrosion caused by chloride was supplied on Page 2 according to the Reviewer’s suggestion.
Comment 11: the combinate of chlorine corrosion and the freeze-thaw cycle
Response 11: We have corrected this error as “combination”.
Comment 12: Explain the reason for this sentence that” Chloride penetration test results show that recovered aggregates can be used in concrete; however, they need to be added with fly ash [17].”
Response 12: Considering the Reviewer’s suggestion, we have added to the texts on the effects of fly ash. We added explanations that “Fly ash, as an industrial by-product, was widely used to improve concrete performance by virtue of its value attributes such as micro-aggregate filling and pozzolanic effect [22-23]. The micro-aggregate filling effect reduced porosity and pozzolanic effect consumed Ca(OH)2, and then ITZ would be enhanced. Finally, concrete performance was enhanced wholly because the permeability of the constituent materials determined the permeability of concrete”. It has been revised in Section 1: Introduction (Page 3).
Comment 13: The cited literature of Kou [18] shows an incomplete conclusion
Response 13: We made a supplementary description of the research background and rewrote this part that “Kou et al. [24] stated that fly ash in proper proportion could improve the frost resistance of recycled concrete and enhance its chloride penetration resistance to some extent. They tested recycled concrete with 0, 25%, and 35% Class F fly ash and pointed out that incorporating 25-35% fly ash was a practical way to improve the chloride permeability resistance at the water-binder ratio of 0.45 and 0.55, which minimized the total charge passed of recycled concrete at 28 and 90 days.”. It has been revised in Section 1: Introduction (Page 3).
Comment 14: The literature cited Kurda [19], which type of concrete? This looks like an overall conclusion but it’s not.
Response 14: We made a supplementary description of the research background and rewrote this part that “Kurda [25] extracted the results of 655 concrete incorporating coarse RCA and FA, at least 28 days, from a lot of papers, it was found that the concrete with 20% fly ash was exhibit-ed a better performance compared to the benchmark concrete.”. It has been revised in Section 1: Introduction (Page 3).
Comment 15: Suggest using Interfacial Transition Zone (ITZ) to explain. “Performance degradation of concrete was related to the interaction between weaker quality aggregate and old mortar adhered to the recycled aggregate [26]. “
Response 15: Description of interface transition zones (ITZ) has been added and adjusted at a more appropriate position. As the following: “Indeed, performance degradation of recycled concrete was due to the presence of two additional interfacial transition zones (ITZ) with low particle density. One of them existed be-tween natural aggregate and old mortar adhered to the recycled aggregate, and the other existed between natural mortar and old mortar [13-14]. The microstructure of ITZ was observed by scanning electron microscope (SEM), the concentration of calcium hydroxide crystal was higher and the porosity was larger[15]. The porosity of the mortar matrix is about 1 / 3 of the maximum porosity of ITZ [16].” It has been revised in Section 1: Introduction (Page 2).
Comment 16: The literature on cited Boukour [19], Boukour manufactured a resinous mortar and indicated that the combined incorporation of 20% rubber waste with 5% and 7.5% brick waste. Is it relevant to your research?
Response 16: The authors carefully read Boukour 's article again and cited relevant data. As the following: “Boukour manufactured mortars with 2.5%, 5%, 7.5%, and 10% brick waste of natural coarse aggregate and there was a gradual downward trend in the 28 days compressive and flexural strengths [32].“ It has been revised in Section 1: Introduction (Page 3).
Comment 17: The reason why fly ash can improve the frost resistance of concrete is not mentioned
Response 17: This explanation was added to Page 3. As the following:“Fly ash, as an industrial by-product, was widely used to improve concrete performance by virtue of its value attributes such as micro-aggregate filling and pozzolanic effect [22-23]. The micro-aggregate filling effect reduced porosity and pozzolanic effect consumed Ca(OH)2, and then ITZ would be enhanced. Finally, concrete performance was enhanced wholly because the permeability of the constituent materials determined the permeability of concrete.” (Section 1: Introduction)
Comment 18: One of the highest disadvantages of recycled aggregate it’s variability in properties, there can be a huge difference between different branches.
Response 18: We also mentioned this issue on Page 4 and it was included that “it has been difficult to unite the physical markers of recycled brick, and dispersion in the findings persists.” Considering the variability of recycled aggregate, we try our best to supplement the test background, the strength of parent concrete, and various physical performance indexes of recycled aggregate. (Section 1: Introduction)
Comment 19: The classification of aggregate particle size is different from international standard.
Response 19: Classification of aggregate size was based on standard ASTM C125. Specifically stated as follows. Coarse aggregate-(1) aggregate predominantly retained on the 4.75mm sieve; or (2) that portion of an aggregate retained on the 4.75mm sieve. Fine aggregate-(1) aggregate passing the 9.5mm sieve and almost entirely passing the 4.75mm sieve and predominantly retained on the 75m sieve; or (2) that portion of an aggregate passing sieve the 4.75mm sieve and retained on the 75m sieve. It has been revised in Section 2.1.2. Natural aggregates and indicated in blue color (Refer Page 4 of the revised manuscript).
Comment 20: What are the criteria followed by the crushing index test.
Response 20: All physical characteristics in Table2 was based on Chinese standard JGJ 52-2006. It should be noted here that “JGJ52-2006” quoted in Section 2.1.3. Recycled aggregate is a well-respected Chinese standard widely used by the construction industry. It has been revised in Section 2.1.3. Recycled aggregates and indicated in blue color (Refer Page 4 of the revised manuscript).
Comment 21: The RBCA mixing ratio was not mentioned in the introduction
Response 21: We added this part of the content to the summary and introduction. It has been revised in Section Abstract, 1: Introduction, and 2.1.3. Recycled aggregates and indicated in blue color (Refer Page 1, 4, and 4 of the revised manuscript, respectively).
Comment 22: What is the difference between figure left and right (Figure 1. Grading curves of RBCA)
Response 22: The vertical coordinates of the left and right figures were respectively classified the individual percentage retained and cumulative percentage retained. The individual percentage retained is the percentage of sieve residue on a certain sieve to the total mass of the sample. The cumulative sieve retained is the percentage referred to all of the sample that is larger than that sieve. We have changed the ordinate of Figure 2 to percent passing (Page 5).
Comment 23: Explain the meaning of HL-HPC, and all commercial codes are to be deleted. Add the value of slump.
Response 23: HL-HPC was the commercial number of a specific batch of the polycarboxylate superplasticizer. In the revised manuscript, the authors have deleted all commercial brands of the fly ash and the superplasticizer used for preparing concrete in this study. The value of slump was supplemented that” to maintain the concrete slump of 90-100mm and slump loss in one hour is 40-50mm.” It has been revised in Section 2.1.4. Other materials and indicated in blue color (Refer Page 5 of the revised manuscript).
Comment 24: Schematic diagram of RBCA used in RAC explained in Figure 2 is incorrect and misleading.
Response 24: We wanted to show the application of recycled aggregate in concrete intuitively by graphics, but it was a pity to be misleading. We have removed this misleading schematic.
Comment 25: Incorrect filling of fly ash proportion in Table 3. Mix proportions (except FA) of recycled concretes (kg/m3). Errors in the writing of symbols in Equation (1)
Response 25: We are very sorry for our negligence and thank the reviewers for finding these mistakes in time. We have corrected it and confirmed it repeatedly. (Page 6)
Comment 26: The procedure for preparing the specimen, the specification for the compressive strength, and how the results of the compressive strength test were obtained.
Response 26: “The preparation process of specimens complied with the standard ASMT C192. The weighed coarse aggregate, partial water, mix the polycarboxylate superplasticizer with the mixing water in advance. Then, start the mixer and add cementitious material and fine aggregate. Mix in the mixer for 3 minutes, rest for 3 minutes, and mix for another 2 minutes to make them mix evenly.” and “Axial compressive strength tests were performed in conformity to ASMT C39 and test results were determined to be the mean values of every 3 specimens.” It has been revised in Section 2.3 Experimental methods and indicated in blue color (Refer Page 6 of the revised manuscript).
Comment 27: Why are the number of freeze-thaw cycles set at 15 and 45 to test the test data.
Response 27: As the Reviewer suggested that the experimental data every five freeze-thaw cycles should be recorded. According to the original plan for the experiment, testing mass loss rate and relative dynamic elastic modulus in the number of freeze-thaw cycles at 5, 10, …, and 45. However, the testing process was affected by the COVID-19 pandemic. We regret that we can’t record this part of the data, which caused a large test interval of the experimental data.
(2.3.1. Freeze-thaw cycles in NaCl solution)
Comment 28: What is a dynamic elastic modulus measurement instrument, is it UPV?
Response 28: It is not the UPV. To determine the elastic modulus of the specimens, the transverse frequency test method adapted from ASTM C215-02, was employed. First, the mass and the average length of the specimens were measured. The specimens were then placed on a 20mm thickness Polystyrene board. A driving unit was attached at the middle of the specimen’s lateral surface, followed by a pickup unit 5mm away from the end of the specimen on the midline of the lateral surface. The transverse frequency was recorded by means of a concrete dynamic elastic modulus testing apparatus. The complete experimental set up is shown in Figure. The pictures of the test equipment (Figure 2) have been supplemented. (2.3.1. Freeze-thaw cycles in NaCl solution)
Comment 29: Incorrect writing of equation 1 about MRL
Response 29: We have corrected it. (Page 6)
Comment 30: Why is the sample depth collected 5cm in the water-soluble chloride test.
Response 30: Referred to ACI 318, concrete protective layer thickness concentrated in 10-50mm, 75mm only in rare cases. Therefore, the chloride ion concentration was focused on in the range of 0-50mm from the erosion surface. It has been revised in Section 2.3.2. Water-Soluble Chloride test and indicated in blue color (Refer Page 7 of the revised manuscript).
Comment 31: Schematic diagram of the procedure for water-soluble chloride content test (Figure 4) has no added value.
Response 31: Water-soluble chloride content test procedures, reference specifications, and solution dosages were detailed added as the following: “Samples were soaked in distilled water, stand for 24h after vibrating for 5 minutes to extract chloride ions. Then 2 drops of phenolphthalein were used as a color indicator and it turned red because the aqueous solution was alkaline. Subsequently, dilute sulfuric acid was added until the solution was colorless. In the end, 10 drops of K2CrO4 solution were an indicator added to record the consumption of AgNO3 when peach precipitation formed. The chemical reactions are shown in Equation (3) and Equation (4) occurred successively. Silver ion of AgNO3 solution reacted with the chloride ion to form silver chloride firstly, and then excess silver ion interacted with chromate ion of K2CrO4 solution to form peach precipitation. Chloride ion content was measured by recording the consumption of the AgNO3 solution.” It has been revised in Section 2.3.2. Water-Soluble Chloride test (Page 7).
Comment 32: How the results of the compressive strength test were obtained, mean values?
Response 32: This content in 2.3 experimental methods was supplemented. “Test results were determined to be the mean values of every 3 specimens.” (Page 6)
Comment 33: The results indicated that RBCA content had a significant effect on mechanical properties and that the 28-day compressive strength of concrete reduced with an increase of RBCA replacement, which was consistent with previous findings by González [30].
Response 33: The authors corrected this vague description and deleted the reference that” The results indicated a negative effect between the content of recycled aggregates and mechanical properties that the 28-day compressive strength of concrete reduced with an increase of RBCA replacement.” It has been revised in Section 3.1.1. Compressive strength and recycled aggregate (Refer Page 8 to 9 of the revised manuscript).
Comment 34: On the other hand, the brick aggregate had a high porosity and water absorption capacity, which allowed it to absorb a large amount of moisture
Response 34: The authors added numbers to support the idea that “water absorption capacity (18%), which was three times than RCA” (Page 9)
Comment 35: Internal curing effect of reclaimed aggregate is only an assumption and a conclusion cannot be inferred.
Response 35: The authors corrected this description that” In addition, the actual water-binder ratio of concrete with RBCA below the designed value results in strength was relatively high.” (Page 9)
Comment 36: Poor discussion of Chapter 3.1.2 of compressive strength and FA
Response 36: The author rewrote this part of the analysis and added trend graphics as Figure 6. “Trends on the strength curve of concrete with different mix ratios exhibited the same. When the content of FA increased from 0%,15% to 30%, these curves went up and then down. Overall, 15% FA generated strength growth to around an average of 7% regardless of the type of recycled aggregate. However, differences in the strength for the concrete produced using 15% FA compared to those of produced no fly ash were negligible. This phenomenon was attributed to micro-aggregate filling and the pozzolanic effects of fly ash [20,21]. Nevertheless, a high volume of fly ash incorporation magnified total porosity result in density reduction.” (Page 9)
Additional alterations made by the authors.
Additional 1: The parent concrete’s strength of recycled concrete aggregate was added: “The compressive strength of parent concretes was 40.7 MPa.” It has been revised in Section 2.1.3. Recycled aggregates and indicated in blue color (Refer Page 4 of the revised manuscript).
Additional 2: Table 2 presents the physical characteristics of recycled aggregates. Water absorption of recycled brick-concrete aggregate, recycled concrete aggregate, and recycled brick aggregate was supplemented, respectively, in Table 2. It has been revised in Section 2.1.3. Recycled aggregates and indicated in blue color (Refer Page 4 of the revised manuscript).
Additional 3: Supplementary content with the water-soluble chloride test procedure. As the following: “Samples were soaked in distilled water, stand for 24h after vibrating for 5 minutes to extract chloride ions. Then 2 drops of phenolphthalein were used as a color indicator and it turned red because the aqueous solution was alkaline. Subsequently, dilute sulfuric acid was added until the solution was colorless. In the end, 10 drops of K2CrO4 solution were an indicator added to record the consumption of AgNO3 when peach precipitation formed. The chemical reactions are shown in Equation (3) and Equation (4) occurred successively. The silver ion of AgNO3 solution reacted with the chloride ion to form silver chloride firstly, and then excess silver ion interacted with chromate ion of K2CrO4 solution to form peach precipitation.” It has been included in Section 2.3.2. Water-Soluble Chloride test indicated in blue color (Refer Page 7 of the revised manuscript, respectively).

Reviewer 3 Report
The article presents an analytical solution for determining the diffusion coefficient of chlorides in concrete, which can take into account the shape of the aggregate. The high accuracy of the analytical solution is confirmed by comparison with experimental data. In the work, the generalized Maxwell approach is extended to determine the diffusion coefficient of chlorides in concrete based on the concrete mesostructure consisting of randomly distributed equivalent elliptical aggregates and cement paste.
The abstract is complete and clear. The review is complete and clear. The manuscript is clear, relevant and well structured. The cited links are relevant, there are few self-citations. The manuscript is scientifically substantiated. Figures and tables are clear, show data correctly, and are easy to interpret. The findings are consistent with the evidence and arguments presented.
Notes:
-
- An extended Maxwell method for macroscopically anisotropic concrete consisting of an equivalent elliptic concrete could be described in more detail.
- How does the porosity of concrete affect (or can affect) the diffusion coefficient of chlorides?
- What is the dependence of the size of aggregate inclusions on the diffusion coefficient of chlorides?
- How difficult is it to extend the proposed method to take into account the noncanonical shape of aggregate inclusions to the diffusion coefficient of chlorides?
- It is desirable that the authors evaluate which parameters of concrete constituents used in the work, in addition to the diffusion coefficient of chlorides, can also significantly affect the calculation of the durability of concrete.
Author Response
The authors would like to thank the reviewer for providing valuable suggestions. Errors in references to tables and figures have been corrected on all pages. Here, we attach the revised manuscript, in the formats of both PDF and MS Word, for your approval. A revised manuscript with the correction sections red marked is attached as the supplemental material and for easy check/editing purpose. We try our best to improve the manuscript and made some changes in the manuscript. We appreciate the Reviewers’ warm work earnestly and hope that the correction will meet with approval.

Round 2
Reviewer 2 Report
No further comments